# Impact of a teacher-led school handwashing program on children's handwashing with soap at school and home in Bihar, India

**James B. Tidwell**[1]*, **Anila Gopalakrishnan**[2], **Arathi Unni**[2], **Esha Sheth**[2], **Aarti Daryanani**[2], **Sanjay Singh**[3], **Myriam Sidibe**[1]

**1** Mossavar-Rahmani Center for Business and Government, Harvard Kennedy School of Government, Cambridge, Massachusetts, United States of America, **2** Unilever PLC, London, United Kingdom, **3** Population Services International, Delhi, India

* ben_tidwell@hks.harvard.edu

**Data Availability Statement:** The data are available on Open Science Framework: http://doi.org/10.17605/OSF.IO/A4R2T.

## Abstract

Handwashing with soap is an important preventive health behavior, and yet promoting this behavior has proven challenging. We report the results of a program that trained teachers to deliver a handwashing with soap behavior change program to children in primary schools in Bihar, India. Ten intervention schools selected along with ten nearby control schools, and intervention schools received the "School of Five" program promoting handwashing with soap using interactive stories, games, and songs, behavioral diaries to encourage habit formation, and public commitment. Households with children aged 8–13 attending the nearby school were enrolled in the study. Handwashing with soap was measured using sticker diaries before eating and after defecation 4 weeks after the intervention was completed. Children in the treatment reported 15.1% more handwashing with soap on key occasions (35.2%) than those in the control group (20.1%) (RR: 1.77, CI: (1.22, 2.58), p = .003). There was no evidence that handwashing with soap after defecation was higher in the treatment group than the control group (RR: 1.18, CI: (0.88, 1.57), p = .265), but there was strong evidence that handwashing with soap was greater in the treatment than in the control before eating (RR: 2.68, 95% CI: (1.43, 5.03), p = .002). Rates of handwashing increased both at home (RR: 1.63, CI: 1.14, 2.32), p = .007) and at school (RR: 4.76, 95% CI: (1.65, 17.9), p = .004), though the impact on handwashing with soap at key occasions in schools was much higher than at home. Promoting handwashing with soap through teachers in schools may be an effective way to achieve behavior change at scale.

## Introduction

Diarrhea and pneumonia are leading causes of illness and death for children under 5 years old [1, 2], as well as for school-aged children [3]. The prevalence of both diarrhea [4] and pneumonia [5] can be reduced significantly by proper handwashing with soap. Though handwashing with soap was once considered one of the most cost-effective interventions to reduce high-

**Funding:** The funder (Unilever PLC) provided support in the form of salaries for authors AG, AU, ES, and AD, but did not have any additional role in the study design, data collection and analysis, decision to publish, or preparation of the manuscript. The specific roles of these authors are articulated in the 'author contributions' section.

burden diseases in lower- and middle-income countries (LMICs) [6], more recent analyses have been more hesitant to make such claims due to variations in the effectiveness of such programs [7], and thus it is very important to understand how such programs work through careful evaluations at small and large scales. A national handwashing program in India based on some of the more effective programs identified to date would lead to an estimated US$5.6 billion annual gain in GDP, a 92-fold return on investment [8], suggesting the importance of taking such interventions to scale.

However, there are many challenges with achieving handwashing behavior change at scale. First, measures used across many studies as proxies for behavior are weak [9], and even the current "gold standard" of structured observation is less than ideal [10]. Second, many programs have focused on health education alone, which is likely insufficient due to the importance of motivated and habitual aspects of behavior [11], which leads to difficulty in establishing the effectiveness of hygiene promotion programs in the aggregate in systematic reviews. Third, even when such reviews carefully examine the available evidence, there are few studies in school contexts in LMICs and most have short follow-up, weak outcome measures, or poor blinding of outcome assessors [12].

Schools are a key setting for the development of effective handwashing promotion programs for many reasons. First, the school environment plays a key role in many health outcomes [13]. Second, many behaviors "track" into adulthood [14], and promoting health in adolescence is a key to reducing health inequities [15]. Third, some behaviors may spread from school-aged children to their parents or younger siblings, especially if accompanied by materials viewed as trustworthy or changes to the home environment initiated by the children [16, 17]. However, programs not specifically designed with this in mind or without accompanying transformative approaches are unlikely to do so [18]. Fourth, many behaviors are setting-specific, and so interventions that exclude schools may be unlikely to lead to behavior change in this important setting [19, 20]. Fifth, the school setting may be more feasible for integrated, sustainable programs that allow delivery at scale, even incorporating related preventive health behaviors such as adding face washing to prevent trachoma to a handwashing program [21], though implementation fidelity is a concern.

However, it is important that programs in such settings are well designed- a review of adolescent health programs found mixed results for school-based behavior change programs, but found that multi-component interventions (combining education with other approaches) were more effective than education alone [22]. Based on this evidence, the World Health Organization has recommended the "Health Promoting Schools," approach, which goes beyond simple education to work more holistically in the school environment [23].

There is limited evidence about the effectiveness of school-based handwashing promotion programs despite the potential to see significant health impacts [24]. Promotional programs may contribute to increasing rates of handwashing with soap when adequate facilities are present [25], especially if they draw attention to the desired behavior [26]. We report here the results of an evaluation of the "School of Five" program, which promoted handwashing with soap via trained teachers in rural Bihar, India, selected because it is likely one of the most difficult settings for such a program. We examine the impacts on handwashing with soap on key public health occasions and differences in impact for school and home settings.

## Methods

This study took place in 20 villages of the Masauhri block of Patna district in Bihar, India. Bihar has the lowest GDP per capita [27], the highest rate of open defecation [28], and the third highest rate of death due to diarrheal diseases [29] of any state in India.

The 4-week campaign trained teachers to deliver the program on hygiene promotion to children in schools based on the "School of Five" curriculum [30] through the NGO Population Services International (PSI). This program was designed by Unilever's health soap Lifebuoy to improve handwashing behaviours at key public health occasions. Intervention components included the presentation of interactive stories, games and songs, the use of animated characters representing occasions for handwashing with soap, the use of visual demonstrations to communicate the presence of germs and how handwashing with soap removes them in comparison to only water, the use of daily diaries by students to record their handwashing behavior, and making a public commitment to handwashing with soap as a group. The program was delivered by government school teachers to whom follow-up mobile messages were sent to confirm the delivery of the curriculum each week. No additional materials were provided to students for use in school or at home, and no provision of soap or alterations to handwashing conditions in homes was made by the program. More detail about the program components is given elsewhere [31].

Ten villages were randomly selected from the 25 where the program was delivered during this phase of the roll out and ten nearby villages similar to these were selected by the evaluation agency as controls. Based on the baseline rates of handwashing, the sample size was adequate to detect a four percentage point difference between intervention and control arms with a 95% confidence and a 80% power for non-clustered data, with the exact percentage change detectible dependent on how clustered any changes might have been at endline. Though the intervention was delivered to the schools, data was collected through household surveys of mothers and their school-age children. Households were selected at random from a school register, and mothers were asked to list any children between 8–13 years old attending the local government school, with a child at random selected for the survey if more than one was eligible in the household. Enrolment took place in July and August 2015, the campaign was delivered for four weeks in September 2015, and follow-up took place in October 2015 with at least 4 weeks between the end of the intervention and follow-up.

The primary outcome of the study was the overall proportion of times that children performed handwashing with soap at key occasions, defined for the study population as after defecation and before eating. These occasions were elicited using sticker diaries [32], where participants recorded a variety of behaviors from the previous day, divided into the three time periods of before, at, and after school, and were then prompted whether and with what they washed their hands before and after key occasions. Sticker diaries were found to be more accurate than some other self-reported measures of handwashing with soap [32], and have the advantage over structured observation that they can capture behavior in a larger time window and in different settings where structured observation may be constrained by logistical, cultural, or privacy considerations.

Log-binomial regression was used to analyze the difference in the primary outcome between the two study arms, with clustering at the individual and school levels. Sub-group analysis was also conducted by type of key occasions and by setting (school vs. home).

Data collection tools were developed by staff members from Nielsen, an independent consumer research agency. The team consisted of supervisors and surveyors, who were trained to understand the objectives of the evaluation and the tools to be used, including mock interviews to become familiar with the tools. Electronic data was stored on password-protected computers, and paper forms were stored in locked cabinets.

Consent was initially sought from head teachers at each school. The market research firm told mothers that they were conducting a study of the day-to-day activities of households in the village and that their information would be kept confidential and only analyzed in the aggregate. Mothers and children were told that their participation was voluntary. Verbal

consent was obtained from each principal, mother, and child due to high levels of illiteracy in the area in keeping with local market research guidelines. The research agency conducting the study followed the procedures and guidelines mandated by the Market Research Society of India [33] and followed standard internal procedures to ensure ethical research standards. An independent research agency collected data for this evaluation. Data received for analysis for this manuscript was fully anonymized. This data was collected initially for company use, and secondary analysis was conducted on anonymized data, and thus the study was considered exempt from the university IRB.

## Results

Basic demographic characteristics of the sample are shown in Table 1. Children surveyed were about evenly split between males and females. Mothers had a mean age of 32 years old, and a majority reported being housewives, though more engaged in agricultural activities in the intervention than in the control. More than half of mothers in the sample were illiterate or had no formal schooling. Due to imbalances between treatment and control mother's profession and education, these were included as covariates in the analysis. Baseline rates of handwashing with soap were indistinguishable in the intervention and control groups.

Children in the treatment reported handwashing with soap on key occasions (35.2%) more than those in the control group (20.1%, RR: 1.77, CI: (1.22, 2.58), p = .003) (Table 2). There was no evidence that handwashing with soap after defecation was higher in the treatment group than the control group (RR: 1.18, CI: (0.88, 1.57), p = .265). However, there was strong evidence that handwashing with soap was greater in the treatment than in the control before eating (RR: 2.68, 95% CI: (1.43, 5.03), p = .002), though it occurred less often than after defecation in both treatment (28.4%) and control (10.0%).

When examining children's handwashing with soap at key occasions separately by setting (i.e. at home vs. at school), there was evidence that rates of handwashing increased both at

**Table 1. Child and parent demographics from treatment and control schools.**

| Variable | treatment | control | p-value |
|---|---|---|---|
| n = | 108 | 117 | |
| Child gender = female (n (%)) | 57 (52.8) | 61 (52.1) | 1.000 |
| Child age (mean (SD)) | 9.9 (1.6) | 10.2 (1.5) | .219 |
| Mother's age (mean (SD)) | 32.4 (7.8) | 32.5 (7.5) | .959 |
| Mother's profession (%) | | | .001 |
| Housewife | 61 (56.5) | 93 (79.5) | |
| Cultivator | 12 (11.1) | 11 (9.4) | |
| Agricultural activities | 34 (31.5) | 13 (11.1) | |
| Other | 1 (0.9) | 0 (0.0) | |
| Mother's education (%) | | | .032 |
| Illiterate | 63 (58.3) | 47 (40.2) | |
| No formal schooling | 11 (10.2) | 9 (7.7) | |
| Up to 4th standard | 15 (13.9) | 25 (21.4) | |
| 5th to 7th standard | 15 (13.9) | 20 (17.1) | |
| 8th to 9th standard | 2 (1.9) | 6 (5.1) | |
| 10th standard | 1 (0.9) | 9 (7.7) | |
| Higher secondary/intermediate | 1 (0.9) | 1 (0.9) | |
| Overall Rate of Handwashing with Soap at Key Occasions | 4.0% (21/529) | 4.1% (24/580) | .892 |

**Table 2. Impact of the School of Five intervention on the percent of key public health occasions where handwashing with soap occurred by occasion and setting.**

| Occasion/Setting | Treatment (% (numerator/ denominator) | Control (% (numerator/ denominator) | Risk Ratio (95% CI) | p-value |
|---|---|---|---|---|
| Overall | 35.2% (161/457) | 20.1% (94/468) | 1.77 (1.22, 2.58) | .003 |
| After defecation | 57.4% (62/108) | 53.2% (58/109) | 1.18 (0.88, 1.57) | .265 |
| Before eating | 28.4% (99/349) | 10.0% (36/359) | 2.68 (1.43, 5.03) | .002 |
| At home | 38.0% (139/366) | 24.1% (90/374) | 1.63 (1.14, 2.32) | .007 |
| In school | 24.2% (22/91) | 4.3% (4/94) | 4.76 (1.65, 17.9) | .004 |

home (RR: 1.63, CI: 1.14, 2.32), p = .007) and at school (RR: 4.76, 95% CI: (1.65, 17.9), p = .004). Handwashing with soap was much less common in the control group in schools (4.3%) than at home (24.2%), and a t-test showed that there was a greater impact on handwashing with soap at key occasions in schools than in the home (RR: 2.70, CI: (1.19, 9.48), p = .025).

## Discussion

The main objective of the school-based handwashing promotion program was to increase handwashing with soap on key public health occasions in the school setting. We find evidence that it succeeded in changing behavior overall, with large impacts before eating (which was very low to begin with) and both in schools and at home among children, though the impact was greater in the school setting than in the home. However, there was no difference between treatment and control arms for rates of handwashing with soap after defecation.

Given that this program was delivered in schools, the higher impact in schools than in the home is not surprising. However, a goal of the program was to impact behavior of children in school, and through them in the home as well, potentially spreading to the rest of the family, which was not found in another evaluation of a similar program delivered by an outside agency (non-teacher led) [31]. Future iterations of the program should pay particular attention to what approaches are most appropriate to address the different settings within which handwashing takes place.

In schools, infrastructure improvements along with triggering social norms may lead to large increases in rates of handwashing with soap [26], but designing effective systems for ensuring on-going provision of soap and water in schools is key. There is good evidence suggesting that school-based programs can drive sustainable behavior change even when delivered with low intensity [34], and so the target rates of handwashing with soap to see the desired public health impact should be weighed carefully against the cost of more intensive, even infrastructure providing interventions.

In homes, many of the behavioral determinants present in targets schools will not be present- social pressures will likely not have changed and infrastructure changes or material provision not made. This may be one of the reasons that structured observation, limited to the home setting, has detected little impact in some school-based hygiene promotion programs [24, 31]. School-based program should consider carefully how to give a role to schoolteachers, incorporate mothers, other family members, school authorities (who may communicate trustworthiness of the messages to those family members [16]), and maybe even whole communities in handwashing promotion programs to see behavior change among key groups such as

children too young to attend school. But, the importance of the school-based setting should also not be ignored in community-based handwashing promotion programs.

This model is likely to be more scalable and sustainable and may have added benefits due to students' pre-existing relationships with and trust in teachers. In addition, mid-day meals are another major government program delivered through schools, and so the synergies of having teachers involved in both activities may have facilitated the observed higher rates of handwashing with soap before meals in school. The reasons for a lack of impact on post-defecation handwashing include that it does not take place at a single point in time, so non-compliance may not be observable to many others, weakening the influence of social norms, in addition to the often poor quality or non-functioning of school toilets. Future programs should also explicitly consider the role of infrastructure and behavioral motives for different key public health occasions for handwashing with soap.

A few key limitations of this study should be noted. First, no data was collected on the available handwashing infrastructure either from mothers or children in homes or from schools (as there was no data collection at all from schools). Second, the sample size was not sufficient to draw implications about specific sub-behaviors of interest. Both of these limit the inferences that can be drawn about, for example, whether increases in handwashing in schools were more driven by those performing it before eating (where peer pressure before a common meal may be strong) or after defecation (which may be less visible to other students) or whether rates of handwashing in the home increased greatly in homes where infrastructure was present, while remaining the same in less-well-equipped homes. Finally, the small number of clusters (schools), despite being controlled for statistically, limits the strength of the evidence as the distribution of rates of handwashing with soap within schools varied greatly.

## Conclusion

Higher rates of handwashing with soap at key public health occasions were found by students in the treatment arm than in the control arm, with larger effects seen in schools and before meals, with increases also found in the home. Due to the importance of setting on behavior, school-based and teacher-implemented programs may be critical to achieving high overall rates of handwashing with soap, making them a key consideration to complement community-based interventions, and there is some impact of school-based programs on student behavior in the home. Learnings from recent trials should be incorporated to develop more effective programs, whose effectiveness at scale should be carefully evaluated in both design and evaluation.

## Acknowledgments

We would like to thank our enumerators, teachers, and school administrative staff for their hard work and dedication to seeing this work reach fruition.

## Author Contributions

**Conceptualization:** Anila Gopalakrishnan, Arathi Unni, Esha Sheth, Aarti Daryanani, Sanjay Singh, Myriam Sidibe.

**Formal analysis:** James B. Tidwell.

**Methodology:** James B. Tidwell.

**Writing – original draft:** James B. Tidwell.

**Writing – review & editing:** Anila Gopalakrishnan, Arathi Unni, Esha Sheth, Aarti Daryanani, Sanjay Singh, Myriam Sidibe.

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
