## [Decision Letter · Decision Letter 0]

30 Dec 2019

PONE-D-19-31072

Impact on handwashing with soap among children at school and home from a teacher-led school handwashing program in Bihar, India

PLOS ONE

Dear Dr Tidwell,

Thank you for submitting your manuscript to PLOS ONE. After careful consideration, we feel that it has merit but does not fully meet PLOS ONE’s publication criteria as it currently stands. Therefore, we invite you to submit a revised version of the manuscript that addresses the points raised during the review process.

Your paper is a very important work. The reviewers have given constructive comments. Kindly go through them carefully and revise. 

We would appreciate receiving your revised manuscript by Feb 13 2020 11:59PM. To enhance the reproducibility of your results, we recommend that if applicable you deposit your laboratory protocols in protocols.io, where a protocol can be assigned its own identifier (DOI) such that it can be cited independently in the future. For instructions see: http://journals.plos.org/plosone/s/submission-guidelines#loc-laboratory-protocols

We look forward to receiving your revised manuscript.

Kind regards,

Vijayaprasad Gopichandran

Academic Editor

PLOS ONE

Journal Requirements:

2. Please ensure that the statistical analyses performed are described in sufficient detail so that the study could be performed again.

"I have read the journal's policy and the authors of this manuscript have the following competing interests: JBT received consulting fees for the data analysis and writing of this manuscript. AG, AU, ES, and AD are employees of Unilever, PLC. JBT had final decision on manuscript content and whether to submit the manuscript for publication."

We note that one or more of the authors have an affiliation to the commercial funders of this research study : Unilever PLC

“The funder provided support in the form of salaries for authors SS, but did not have any additional role in the study design, data collection and analysis, decision to publish, or preparation of the manuscript. The specific roles of these authors are articulated in the ‘author contributions’ section.”

Reviewers' comments:

Reviewer's Responses to Questions

**Comments to the Author**

1. Is the manuscript technically sound, and do the data support the conclusions?

Reviewer #1: Yes

Reviewer #2: Partly

2. Has the statistical analysis been performed appropriately and rigorously? 

Reviewer #1: No

Reviewer #2: No

3. Have the authors made all data underlying the findings in their manuscript fully available?

Reviewer #1: No

Reviewer #2: No

4. Is the manuscript presented in an intelligible fashion and written in standard English?

Reviewer #1: Yes

Reviewer #2: No

5. Review Comments to the Author

Reviewer #1: First of all, I will like to congratulate the authors for doing such a nice research in a deprived area of Patna district of Bihar. But the manuscript have some issues which need to be corrected..

1. Title of the tables should be full not a fragment of sentence.. The tables should be self explanatory.

2. There was no statistics employed in between demographic characteristics of control and treatment group for successful matching.

3. In table 2, 'n' is changing for every variable thus authors should mention 'n' for each variable.

4. Do the students were supplied with soaps during the study?

5. There was no data on baseline hand washing practices of the students.

6. In RR estimation do the researchers adjusted for clustering effect or it is without adjustment

Reviewer #2: Dear Authors, I appreciate your efforts in choosing the subject of great importance in today’s context for India.

However, I have reservations for its publications. The reasons for the same is enumerated below:

1. I am convinced with the background on the need of the study. However, there is limited explanation of “school of five programme”- the type, format, frequency, duration, modules used for training and whether the training model was piloted in any other areas before its implementation as an intervention. If yes, what were the challenges and operational issues faced by the investigators during the pilot and its validation. It is also not clear whether the focus of training programme was to have the behavioral changes among the students or mothers or both.

2. The research question is not clear as it states that they are reporting the evaluation done for the programme and at the same time it is said they want to measure the impact of the programme. Probably, they are interested in comparing the differences between the intervention and non-intervention arm. The measure of outcome is missing in the research question.

3. It is less evident whether there could be any behavior change in the context of this settings whether it is appropriate to have an impact study with one month of implementation.

4. The sampling of the villages and the schools explained in the methods are incomplete. The sampling frame, the sampling technique, the sample size calculation, the assumptions made to extrapolate the findings to the general population are inadequately explained.

5. The information on the methods of measuring the outcomes at school and the mother at the home settings are not explained. Hence, it is to difficult to establish the findings of the authors presumed to have captured.

6. The change of behavior among any person in any settings is dependent on many factors. For this study, the authors have relied on few variables like mother’s education and profession. However, there are many other important key factors like the type of housing, socio-economic status of the family, the type of school, the father’s education and profession, the siblings in their house, their exposure to social media etc., which were ignored in this study.

6. PLOS authors have the option to publish the peer review history of their article (what does this mean?). If published, this will include your full peer review and any attached files.

Reviewer #1: No

Reviewer #2: Yes: Sharath Burugina Nagaraja

---

## [Author Response · Author response to Decision Letter 0]

10 Feb 2020

We have attached an easier version of the response to reviewers using color-coding for our responses/amendments, but are also pasting the text here as well if needed by the system: 

We thank the editor for the chance to respond to reviewer comments and amend the manuscript to address their concerns. Our responses below are in blue, text from the manuscript in blue italics, and text we’ve added to the manuscript in bolded blue italics below.

5. Review Comments to the Author

Reviewer #1: First of all, I will like to congratulate the authors for doing such a nice research in a deprived area of Patna district of Bihar. But the manuscript have some issues which need to be corrected.

1. Title of the tables should be full not a fragment of sentence. The tables should be self-explanatory.

We have expanded the table descriptions as follows:

Table 1: Child and Parent Demographics from Treatment and Control Schools

Table 2: Impact of the School of Five Intervention on the percent of key public health occasions where handwashing with soap occurred by occasion and setting

2. There was no statistics employed in between demographic characteristics of control and treatment group for successful matching.

We have added p-values associated with t- and chi-squared-tests as appropriate, and updated the description to address imbalances and inclusions in the proceeding analysis:

Variable treatment control p-value

n= 108 117 

Child gender = female (n (%)) 57 (52.8) 61 (52.1) 1.000

Child age (mean (SD)) 9.9 (1.6) 10.2 (1.5) .219

Mother's age (mean (SD)) 32.4 (7.8) 32.5 (7.5) .959

Mother's profession (%) .001

 Housewife 61 (56.5) 93 (79.5) 

 Cultivator 12 (11.1) 11 (9.4) 

 Agricultural activities 34 (31.5) 13 (11.1) 

 Other 1 (0.9) 0 (0.0) 

Mother's education (%) .032

 Illiterate 63 (58.3) 47 (40.2) 

 No formal schooling 11 (10.2) 9 (7.7) 

 Up to 4th standard 15 (13.9) 25 (21.4) 

 5th to 7th standard 15 (13.9) 20 (17.1) 

 8th to 9th standard 2 (1.9) 6 (5.1) 

 10th standard 1 (0.9) 9 (7.7) 

 Higher secondary/intermediate 1 (0.9) 1 (0.9) 

Due to imbalances between treatment and control mother’s profession and education, these were included as covariates in the analysis.

We have also updated Table 2 to reflect these statistical controls as follows:

Occasion/Setting Treatment 

(% (numerator/

denominator) Control

(% (numerator/ denominator) Risk Ratio

(95% CI) p-value

Overall 35.2%

(161/457) 20.1%

(94/468) 1.77

(1.22, 2.58) .003

After defecation 57.4%

(62/108) 53.2%

(58/109) 1.18

(0.88, 1.57) .265

Before eating 28.4%

(99/349) 10.0%

(36/359) 2.68

(1.43, 5.03) .002

At home 38.0%

(139/366) 24.1%

(90/374) 1.63

(1.14, 2.32) .007

In school 24.2%

(22/91) 4.3%

(4/94) 4.76

(1.65, 17.9) .004

3. In table 2, 'n' is changing for every variable thus authors should mention 'n' for each variable.

The n is changing for each variable and between treatment and control, and thus we have indicated % (numerator/denominator). We have amended the initial column to make this clear. 

4. Do the students were supplied with soaps during the study?

No- we have added the following description to the methods section:

No additional materials were provided to students for use in school or at home, and no provision of soap or alterations to handwashing conditions in homes was made by the program.

5. There was no data on baseline hand washing practices of the students.

Since the analysis was not performed using a difference-in-differences approach, the baseline handwashing rates were not included, as is standard in a randomized trial. However, due to the clustered nature of the data by school, we have added a report of the baseline rates of handwashing into Table 1.

Overall Rate of Handwashing with Soap at Key Occasions 4.0% (21/529) 4.1%

(24/580) .892

And the following text was added to describe the result:

Baseline rates of handwashing with soap were indistinguishable in the intervention and control groups. 

6. In RR estimation do the researchers adjusted for clustering effect or it is without adjustment

The methods section describes that clustering was done at the individual and school levels.

Reviewer #2: Dear Authors, I appreciate your efforts in choosing the subject of great importance in today’s context for India.

However, I have reservations for its publications. The reasons for the same is enumerated below:

1. I am convinced with the background on the need of the study. However, there is limited explanation of “school of five programme”- the type, format, frequency, duration, modules used for training and whether the training model was piloted in any other areas before its implementation as an intervention. If yes, what were the challenges and operational issues faced by the investigators during the pilot and its validation. It is also not clear whether the focus of training programme was to have the behavioral changes among the students or mothers or both.

The program has been deployed in many countries for a number of years and extensive documentation is given in the provided website. We have provided an additional reference to where the behavioral components are described in detail in another publication. 

More detail about the program components is given elsewhere (Lewis, Greenland, Curtis, & Schmidt, 2018).

At no point in the paper is there a reference to anything other than behavior change of children. 

2. The research question is not clear as it states that they are reporting the evaluation done for the programme and at the same time it is said they want to measure the impact of the programme. Probably, they are interested in comparing the differences between the intervention and non-intervention arm. The measure of outcome is missing in the research question.

This comment is unclear to us. We were evaluating the impact of the program. The primary outcome is described in the methods section as (with an additional clarification that these are children’s’ outcomes added), “The primary outcome of the study was the overall proportion of times that children performed handwashing with soap at key occasions occurred, defined for the study population as after defecation and before eating.”

3. It is less evident whether there could be any behavior change in the context of this settings whether it is appropriate to have an impact study with one month of implementation.

Again, this comment is unclear to us. There certainly was behavior change within one month of implementation. The sustainability of such behavior change was beyond the scope of this study.

4. The sampling of the villages and the schools explained in the methods are incomplete. The sampling frame, the sampling technique, the sample size calculation, the assumptions made to extrapolate the findings to the general population are inadequately explained.

The study setting, sampling frame, and sampling technique are given in the methods section. We have added a note on the sample size as follows: 

Based on the baseline rates of handwashing, the sample size was adequate to detect a four percentage point difference between intervention and control arms with a 95% confidence and a 80% power for non-clustered data, with the exact percentage change detectible dependent on how clustered any changes might have been at endline. 

It is not clear to us what is meant by “the assumptions made to extrapolate the findings to the general population are inadequately explained.” The study findings are certainly relevant for the study setting, and generalizability is limited by how similar to this setting another population is along many different dimensions. We do not feel like a generic statement about it being unclear how applicable these findings are to other settings adds anything that is not implicit in any such study.

5. The information on the methods of measuring the outcomes at school and the mother at the home settings are not explained. Hence, it is too difficult to establish the findings of the authors presumed to have captured.

Outcomes were not assessed for mothers. Sticker diaries are described and referenced as the tools for eliciting the behavior of children in the methods section.

6. The change of behavior among any person in any settings is dependent on many factors. For this study, the authors have relied on few variables like mother’s education and profession. However, there are many other important key factors like the type of housing, socio-economic status of the family, the type of school, the father’s education and profession, the siblings in their house, their exposure to social media etc., which were ignored in this study.

Additional demographic controls may have been helpful in conducting the analysis. However, we feel that having explored in this revision the baseline comparability of the rates of handwashing, the similarity across children’s demographic factors, incorporating statistical controls for differences in mothers, and the explicit acknowledgement of the lack of household infrastructure data in the limitations section, we have adequately addressed these concerns given the limited strength of the claims made about the level of evidence and the clear differences observed between intervention and control and across settings for the primary outcomes.

---

## [Editor Report · Decision Letter 1]

12 Feb 2020

Impact of a teacher-led school handwashing program on children’s handwashing with soap at school and home in Bihar, India

PONE-D-19-31072R1

Dear Dr. Tidwell,

We are pleased to inform you that your manuscript has been judged scientifically suitable for publication and will be formally accepted for publication once it complies with all outstanding technical requirements.

With kind regards,

Vijayaprasad Gopichandran

Academic Editor

PLOS ONE
---

## [Editor Report · Acceptance letter]

18 Feb 2020

PONE-D-19-31072R1 

Impact of a teacher-led school handwashing program on children’s handwashing with soap at school and home in Bihar, India 

Dear Dr. Tidwell:

I am pleased to inform you that your manuscript has been deemed suitable for publication in PLOS ONE. Congratulations! Your manuscript is now with our production department. 

With kind regards,

on behalf of

Dr. Vijayaprasad Gopichandran 

Academic Editor

PLOS ONE